# Tumor-to-Tumor Metastasis of Lung Cancer to Kidney Cancer: A Review of the Literature and Our Experience

**DOI:** 10.3390/diagnostics14050553

**Published:** 2024-03-05

**Authors:** Catalin Baston, Andreea Ioana Parosanu, Mihaela Mihai, Oana Moldoveanu, Ioana Miruna Stanciu, Cornelia Nitipir

**Affiliations:** 1Faculty of Medicine, Carol Davila University of Medicine and Pharmacy, 8 Sanitary Heroes Boulevard, 050474 Bucharest, Romania; catalin.baston@umfcd.ro (C.B.); oana.moldoveanu@drd.umfcd.ro (O.M.); cornelia.nitipir@umfcd.ro (C.N.); 2Fundeni Clinical Institute, 022328 Bucharest, Romania; mictekro@gmail.com; 3Elias University Emergency Hospital, 011461 Bucharest, Romania

**Keywords:** tumor-to-tumor metastasis, kidney cancer, lung adenocarcinoma, immunohistochemistry

## Abstract

Tumor-to-tumor metastasis (TTM) is a rare phenomenon documented in patients with multiple primary cancers. This condition is defined as a metastasis between two true primary tumors. The most frequently reported recipient tumor is renal cell carcinoma (RCC), and the lung carcinomas are the most common metastatic tumor donors. Therefore, this paper attempts to address the current gap in knowledge about this rare phenomenon. The first part of this review outlines the recently proposed models and mechanisms involved in the TTM process. The second part then summarizes and analyzes previous case reports in the literature. We also present our experience with the case of lung cancer that metastasized into RCC. Given the sporadic incidence of TTM, no specific management guidelines exist. Therefore, considering TTM in patients with multiple primary tumors is important as it could potentially modify the oncological management offered.

## 1. Introduction

Lung cancer is the second most common cancer worldwide and has one of the most unfavorable prognoses, which is reflected in a low relative 5-year survival rate of about 20%. Distant metastasis is the main cause of death in patients with pulmonary cancer, and over 50% of patients with lung cancer already have metastases when they are first diagnosed. Despite significant advances in the prevention, detection, and treatment of lung cancer, it remains the leading cause of cancer-related deaths [1,2,3,4].

Renal cell carcinoma, on the other hand, is the most common type of kidney cancer in adults, accounting for approximately 3% of all cancers. The 2022 World Health Organization Classification recognizes more than 14 distinct histologic subtypes of RCCs. However, the most common histological subtypes known for decades include clear-cell RCCs (ccRCCs), papillary RCCs (pRCCs), and chromophobe RCCs (crRCCs) [5,6].

Thus, the probability of both cancers coexisting is not neglectable. Moreover, tumor-to-tumor metastasis, defined as a metastasis from one primary cancer to another, is an unexpected event, identified in less than 1% of cancer patients during autopsy [7]. However, TTM is not as rare as previously thought. Since Berent’s original case report in 1902, almost 200 cases have been reported in the literature [8,9,10,11,12]. The most frequently reported recipient tumor is RCC, and lung carcinomas are the most common metastatic tumor donors. Given the sporadic incidence of TTM, no specific management guidelines exist [13].

## 2. The Defining Hallmark of Metastasis

Metastases are the hallmarks of cancer that are responsible for about 90% of cancer deaths [14,15,16]. The term metastasis was coined nearly 200 years ago by the gynaecologist Joseph Recamier who described the invasion of the bloodstream by cancer cells [17]. This phenomenon involves the spread of cancer cells from the original organ to distant lymph nodes or organs. However, the molecular mechanisms of metastasis are complex and not well understood. Metastases develop following evolutionary processes during which tumor cells escape immune recognition and disseminate from the primary site through the bloodstream or lymphatic system to other organs. Cancer invasion and metastasis depend on several factors, which include the loss of adhesion molecules, such as E-cadherin, and weakening of cell–cell adhesion; thus, the expression of pro-angiogenic cytokines, such as vascular endothelial growth factor (VEGF), is increased. Furthermore, increased angiogenesis contributes to tumor growth by increasing the supply of nutrients to tumor cells and facilitating motility and invasiveness [18,19,20,21,22,23].

Several characteristics favor the spread of metastases, including the anatomical site; tumor volume; tumor vascularity, defined by increased microvessel area; and perivascular or perineural invasion. In addition, both local and systemic immunosuppressive factors are known to promote metastasis [24,25,26,27,28,29,30,31,32,33]. However, it is important to differentiate metastases from multiple primary tumors, which is the occurrence of a second primary malignancy in the same patient.

The diagnosis of multiple synchronous or metachronous primary malignancies has been reported rather frequently during the past decade. Their incidence varies between 2.4% and 17% of all cancers [34,35,36,37]. Moreover, patients with previously diagnosed carcinoma have a 20% higher risk of developing new primary cancer compared with the general population [38,39].

## 3. Principles of Tumor to Tumor Metastasis

Although two or more cancers occurring together in the same individual is relatively common, multiple primary cancers occurring in the same organ is a rare clinicopathological entity. TTM and collision tumor are two rare metastatic processes that are not always easy to differentiate and are most likely to be confused.

TTM is a rare phenomenon in which one tumor metastasizes to another unrelated primary tumor. This rare phenomenon was first described in 1902 by W. Berent, who reported a case of metastasis from squamous cell carcinoma of the jaw to RCC [8]. Later, in 1968, L.V. Campbell proposed a set of diagnostic criteria for the diagnosis of TTM [40]. These criteria have been validated by numerous case reports during the last 50 years and include the following: (1) the presence of more than two primary cancers, of which (2) the host or the recipient tumor must be a true benign or malignant neoplasm, with the exception of lymphomas or leukemia, and (3) the metastatic or donor tumor must develop in juxtaposition to the recipient tumor with areas of intermingling [41,42,43,44,45,46]. Additionally, according to Han H.S. et al., the diagnosis of TTM can be truly supported when the following conditions are met: (1) the metastatic lesion must be partially surrounded by the host tumor tissue, and (2) this metastasis must be proven to have originated from another primary cancer [47]. Most recently, in 2017, Richter et al. proposed two main criteria to differentiate tumor-to-tumor metastasis from collision tumor which include the presence of tumor foci with distinct histology in the host tumor and the presence of a metastasized primary cancer (Table 1) [48].

On the contrary, there is another phenomenon, called collision tumor, which is a true coexistence of two neoplastic lesions that maintain distinct borders [49]. Brandwein-Gensier and his colleagues provided a hypothesis to explain the collision tumor phenomenon. According to their theory, it is possible for two independent clonal cancer cells with entirely different genetic phenotypes and histological differentiation to grow in proximity to each other, leading to the origin of a polyclonal tumor [50].

Moreover, these tumors must be distinguished from mixed or pleomorphic tumors, which are characterized by an admixture of cells with a common origin, for example, mixed epithelial and stromal tumors of the kidney or mixed chromophobe and clear-cell type of RCC, when the IHC evidences an imperfect staining pattern of both chromophobe and clear cells in the same tumor [51,52,53,54,55].

### 3.1. What Are the Underlying Mechanisms for TTM?

Currently, two independent theories of Paget and Ewing, respectively, explain the complex mechanisms of metastasis development. In 1889, the English surgeon Stephan Paget proposed the seed and soil hypothesis after evaluating the autopsy records of 735 breast cancer patients and concluding that metastasis did not occur at random [56]. The ability of cancer cells (the ‘seeds’) to grow in a particular organ or tissue (the ‘soil’) is termed ‘organotropism’, and is improved when the right conditions are present (Figure 1). The author further hypothesized that the tumor microenvironment is crucial for tumor growth and spread. Strong evidence suggests that tumor cells selectively settle in and proliferate within particular organs [57,58,59,60]. Kinsey was the first to report an experimental study of preferential metastasis. The author injected S91 Cloudman melanoma cells with a consistent preference for pulmonary tissue into small grafts of different organs that were implanted into the thighs of mice and demonstrated that tumor cells metastasized to normal and ectopically pulmonary grafts, but not to any other tissues [61].

Despite Paget’s seed and soil hypothesis being an appealing metaphor, the theory was not unanimously accepted. In 1928, the American pathologist James Ewing proposed the mechanical theory, highlighting that metastases can be explained using purely mechanical and circulatory forces (Figure 2) [62]. Paget’s theory was based on Virchow’s hypothesis, which considered metastasis to be the consequence of the arrest of tumor cell emboli in the vascular system. Consequently, tumor cells metastasize to a specific organ through the bloodstream or lymphatic system, and the first tissue encountered by tumor cells is the organ with the highest number of metastatic colonies [63,64,65,66,67,68].

From a broad perspective, both hypotheses are equally correct. Ewing’s theory explains the anatomical migration of tumor cells to locoregional lymph nodes and distant organs, and Paget’s theory explains specific metastasis in certain organs.

### 3.2. Why Is Metastasis of Cancer into Cancer Uncommon?

Fifty years ago, Rabson et al. suggested that tumor cells may produce substances locally antagonistic to the second malignancy [43]. Thus, the development of a new tumor is inhibited by the presence of another tumor [69,70]. Kikuchi et al. argued the differences between normal and cancerous cells and experimentally confirmed that neoplastic tissue is an unsuitable soil for the growth of another cancer [71]. Furthermore, the nutritional requirement for the growth of a tumor is high and can compete with other tumor growth [44]. However, this phenomenon of TTM, although infrequent, is not as rare as previously believed. To our knowledge, almost 150 cases of TTM have been reported in the literature [72,73,74].

Any benign or malignant lesion can be the recipient tumor. The three most important characteristics of an ideal recipient tumor are as follows: it should be richly nourished to increase donor tumor growth; a hypervascular organ is located close to the blood vessels, favoring hematogenous metastasis; and it should have a slower tumor growth rate to permit donor tumor cell proliferation [75].

Meningiomas are the most common donors of TTM. On the other hand, RCC is the most common recipient of TTM. This was first noticed in 1969 by Campbell et al. in a series of 22 cases, where the RCC was the leading recipient with a rate of 65% of all cases [40].

This phenomenon is explained by the fact that kidneys are known to have significant hypervascularity, receiving over 25% of circulating blood volume, providing a nutrient-rich environment for pulmonary metastases. Kidneys also contain cell adhesion molecules, such as cadherins, implicated in various biological processes, which may explain the higher rates of metastasis. Moreover, ccRCCs are characterized by mutations in the VHL tumor suppressor gene, which contribute to the constitutive expression of hypoxia-inducible factors 1 and 2 alpha and increased serum levels of VEGF. As a result, ccRCCs are highly vascularized tumors, providing a microenvironment with abundant micronutrients favorable to TTM [76,77,78,79].

However, the donor tumor is usually a malignant lesion because of its high aggressiveness. Lung and breast carcinomas are most consistently the metastatic tumor donors. Few reviews have previously reported that lung cancer is the primary tumor that metastasizes most frequently in other neoplasms, representing up to 50% of all TTM donors. The explanation is given by the increased incidence of lung cancer in the general population and its high capacity for metastatic dissemination. Among pulmonary cancers, adenocarcinoma is the most prevalent lung cancer subtype (67%), followed by squamous cell carcinoma (27%) and small-cell lung cancer (SCLC) (6%). Less common metastatic tumor donors have been reported, including the kidney, skin, prostate, ovary and cervix [80,81,82,83,84].

### 3.3. Does Anatomical Proximity Play a Role in TTM?

To explore the spread preference of cancer metastases, Jiang and colleagues constructed a cancer metastasis spread diagram. With the exception of locoregional lymph nodes, the most common sites of metastases were the liver, bones, lungs and brain. The authors also identified a group of organs and tissues with tropisms for lung tumor cells, such as head and neck, skin, thyroid and kidney, most of which are located close to the lung [85].

These data are supported by a few reports suggesting that lung tumors show a preference for metastasis in certain host tumors [86]. An extensive review of the English literature of 80 cases of TTM from lung carcinoma found RCCs to be the most common host tumors (25%), followed by adrenocortical adenomas (22.5%), meningiomas (20%), thyroid carcinomas (5%) and intracranial schwannomas (5%) [87].

However, there are few case reports of TTM from lung cancer to genitourinary and digestive cancers, such as prostatic, gastric and colorectal carcinoma. Therefore, the TTM-spreading patterns still need further investigation [88,89,90].

## 4. Review of Available Evidence on Metastatic Lung Cancer within Renal Cell Carcinoma

### 4.1. Review Approach and Methods

To date, approximately 150 cases of TTM have been reported in the literature [72,73,74]. However, renal cell carcinoma is the most frequently reported recipient tumor, and lung carcinoma is the most common metastatic tumor donor.

In order to identify studies that report lung cancer metastasis to RCC, we conducted an extensive literature search using the best available resources. We scoured the online repositories of PubMed-MEDLINE from their inception to December 2023 (Figure 3). Our search query line was carefully crafted to ensure we found every relevant piece of literature. The line was as follows: ((“tumour-to-tumour metastasis” [All Fields]) OR (“kidney cancer”) OR (“lung cancer” [All Fields])).

Our search identified 133 studies from PubMed. The inclusion criteria were case reports, case series or original research. A total of 37 studies dealing with other topics, such as neuroimaging, pathology or other anatomic localizations, were excluded; the last screening examined the full texts. Finally, we included eight articles in our review (Table 2) [91,92,93,94,95,96,97,98].

Unfortunately, certain limitations were encountered during this study, including constraints related to the database and keywords, potential language biases, date range limitations, selection criteria, and publication bias. Additionally, the challenges inherent in manual search methods may have restricted the inclusion of all relevant studies.

### 4.2. Results

TTM represents a challenge for differential diagnosis and management. The majority of TTM cases might be clinically subdiagnosed, and most cases are discovered during autopsy. Recently, Sawada T. et al. presented a 97-year-old patient with features of myocardial ischemia who died of sudden cardiac arrest. During the autopsy, a metastatic adenocarcinoma of the lung to RCC was documented [92]. Furthermore, Matsukuma S. et al. investigated autopsy reports of 47 cases of lung tumors and found that the incidence of TTM to RCC was the highest (83%) [91]. Sella A. et al. evaluated autopsy reports of 1136 oncological cases from the M.D. Anderson Hospital and Cancer Institute and found that TTM may not be as rare as the literature describes. Of the 61 cases of TTM identified, 3 RCC metastases were due to a lung adenocarcinoma serving as the donor tumor [97].

Importantly, conventional imaging tests, such as CT scans, are considered insufficient for accurate TTM detection. Rais G. et al. reported that FDG-PET/CT can provide reliable diagnostic information about TTM and serve to guide a biopsy procedure for a specific metastatic lesion, even if most cases were discovered incidentally in living patients or during autopsy. Furthermore, the authors reported the first case of an incidental PET/CT finding consistent with a TTM from lung cancer to RCC [93].

Regarding lung cancer as a donor, adenocarcinoma histology was the most frequently reported. Reflecting on the aggressiveness of the original tumor, most lung cancers were widespread, with other visceral metastases, including lymph nodes and bone metastasis. Here, we report the first case of TTM from lung carcinoma to ccRCC, with widespread pleural and brain metastasis.

In addition to adenocarcinoma, other lung cancer histologies were associated with TTM. Two cases of squamous cell lung carcinoma were reported previously in the literature [93,95]. Furthermore, we found two cases of neuroendocrine lung carcinoma metastasis in a ccRCC [91,96].

According to the existing literature, any benign or malignant neoplasm can be the recipient of TTM. Wu P.S. et al. reported a rare case of lung adenocarcinoma metastasizing into a kidney angiomyolipoma; however, given that angiomyolipoma accounts for less than 1% of surgically removed renal tumors, it is a less common recipient compared with RCC.

Regarding benign kidney tumors, oncocytomas are the most common benign solid kidney neoplasms that account for 3–7% of all solid renal masses [94]. Altinok G. reported one rare case of renal oncocytoma harboring metastatic squamous cell carcinoma of the lung. Radiologically, in the two cases of benign kidney tumor recipients, there was no evidence of other systemic dissemination [95].

Importantly, TTM was not associated with age, tumor size or the presence of other metastases. The median age at diagnosis was 68 years, with a minimum of 48 years and a maximum of 88 years. However, the number of male patients was higher, with a sex ratio of 3:1.

A comparison with previous reports revealed that, although the receptor tumors vary widely in size, from 1.2 to 10 cm, TTM were commonly found to be small tumors of 4 cm or less. Our findings were, however, contradictory. In this case, the tumor was even asymptomatic and was found to occupy almost the entire right kidney, with a diameter of 9/5.5 cm. As for the clinical course, the majority of patients were asymptomatic, and the diagnosis was made incidentally or during an autopsy. Among other coexisting tumors, three patients presented with visceral metastasis. Distant metastases typically involve the lung, liver and lymph nodes. However, bone metastases were most frequently reported. As we have mentioned earlier, this was the first case reported with pleural and cerebral metastases. This case emphasizes the importance of screening for asymptomatic brain metastases during advanced cancer stages. Unfortunately, brain metastases are diagnosed in up to 30% of patients with lung adenocarcinoma and between 3 and 17% of patients with ccRCC. Moreover, 5% of patients with metastatic kidney cancer and 28% of those with metastatic NSCLC may harbor metastatic occult brain disease [99,100,101,102,103,104]. These data once again underline the importance of whole-body imaging staging.

Regarding the morphological features and immunohistochemical findings, donor lung adenocarcinomas revealed positive staining for Napsin A, TTF-1 and vimentine.

IHC was also required for the subtyping of non-small-cell carcinoma. Hence, p40+ and CK5/6 positivity helped to distinguish squamous cell carcinomas from adenocarcinomas of the lung. On the other hand, SCLC stained most frequently with epithelial markers. There were also two cases of SCLC identified with positive CK7, TTF-1, chromogranin, CD56 and synatopsin.

Even though there are no specific immunohistochemical markers of ccRCC, the typical immunophenotype includes strong staining for CAIX, cadherin, vimentine and CD117. RCCs were generally CK7-negative, with the exception of papillary RCC. Smooth muscle actin and HMB-45 positivity helped define renal angiomyolipoma, and the immunohistochemical profile of renal oncocytoma showed positivity for CD117 and E-cadherin.

After reviewing the main principles of TTM and providing a literature review, we present a case report highlighting this particular entity’s unique clinical features and diagnosing challenges. The diagnosis of TTM relies on histological examination because it does not have imaging or macroscopic morphological peculiarities.

### 4.3. Case Report: Tumor-to-Tumor Metastasis to a RCC as the Initial Presentation of a Lung Adenocarcinoma

In May 2023, a 74-year-old man was admitted to the hospital with a 2-month history of progressive dyspnea, cough and weight loss. The patient had no fever or chest tightness. He used to be in good physical shape and had no history of major illnesses.

Physical examination revealed dullness to percussion and reduced breath sounds over the left lung field. The patient showed no signs of respiratory distress, despite a significantly decreased peripheral O_2_ saturation of 88% while breathing ambient air. Other vital signs were normal (blood pressure 132/77 mm Hg, heart rate 81 beats/min and respiratory rate 19/min).

A chest X-ray revealed extensive opacification of the left hemithorax. On biological assessment, hematology, electrolytes and liver and kidney function markers were all normal aside from moderate anemia. During additional workup, enhanced thoracic computed tomography (CT) was performed, which confirmed left pleural effusion. A CT scan also revealed a smooth thickening of the left posteromedial parietal pleura with nodular components, with a maximum diameter of 13/20 mm and a few multifocal, multilobar and peripherally infracentimetric nodules (Figure 4). A right renal tumor was incidentally discovered at the level of the abdominal segment that was partially included in the scanned field. A subsequent contrast-enhanced CT scan of the abdomen and pelvis revealed a well-defined, heterogeneous large mass measuring 12.5 × 9 cm that originated from the upper pole of the right kidney, with cystic, hemorrhagic and necrotic components (Figure 5).

Thoracentesis was performed to diagnose and treat pleural effusions. Cytological examination of pleural fluids revealed classical features of malignancy, with marked hypercellularity, significant cell atypia with large eccentric nuclei and vacuolated cytoplasm. Immunohistochemistry (IHC) staining of the pleural fluid sample revealed diffuse cytoplasmic positivity for thyroid transcription factor-1 (TTF-1), Napsin-A and cytokeratin 7 (CK7).

The patient subsequently underwent radical nephrectomy through a left flank intercostal incision, with the removal of the right kidney and the surrounding fatty tissue. The adrenal gland and lymph nodes around the kidney were also removed. Thus, the patient was discharged on postoperative day 7 without any complications.

Pathological examination revealed distortion of the kidney due to the presence of a 10.5/7.5/5.5 cm mass that extended into the perinephric space. The mass also showed variable degrees of hemorrhage, necrosis and cystic degeneration, but without invasion of the adrenal gland or regional lymph nodes. Macroscopically, the tumor consists of an admixture of solid and cystic components with variable degrees of internal necrosis. Microscopically, the tumor cells were polygonal and spindle-shaped, with clear and eosinophilic cytoplasm from compact nests and sheets with intervening blood vessels. These areas are surrounded by large neoplastic cells with nuclear pleomorphism and hyperchromasia organized in a glandular structure with tubulocystic and papillary patterns, as well as mucin production. Immunohistochemically, the tumor showed two unique histologic patterns. Some areas of polygonal to round cells with clear cytoplasm were diffusely and intensively positive for carbonic anhydrase IX (CAIX) and paired-box gene 8 (PAX8), with the absence of staining for CK7, CK20, TTF-1 and caudal-type homeobox 2 (CDX2). These markers represent the characteristic immune profile of ccRCC. Other areas of carcinoma displayed staining patterns of pulmonary adenocarcinoma with strong TTF-1 and CK7 positivity while being negative for CK20. These cells were also negative for NK3 homeobox 1 (NKX3), CDX2, PAX8, and CAIX, confirming the tumor to be of lung origin (Figure 6).

Thus, IHC played a critical role in establishing the diagnosis of a TTM. The final pathology demonstrated a lung ADK metastasis within a ccRCC.

The patient was referred to an oncologist for further treatment. Unfortunately, 14 days after surgery, the patient presented with new-onset neurological symptoms, including headaches, lower limb weakness and seizures. Multiple cerebral metastases were visible on a contrast-enhanced MRI, with a significant mass effect. Surgery was not a reasonable option, and whole-brain radiation therapy was indicated. The patient exhibited a very rapid disease course; his condition progressively worsened, and he died six weeks later.

## 5. Conclusions

In summary, we aimed to raise awareness and highlight some issues related to the pathology and diagnosis of TTM. Pulmonary carcinoma metastatic to RCC represents the most frequent presentation of a TTM phenomenon. Most of the cases are manageable but not curable, and a consensus about TTM definition, diagnosis and treatment is lacking. Diagnosing TTM disease in the kidney requires a high level of suspicion. It is difficult to differentiate between RCC and renal metastasis on imaging. Therefore, the immunohistochemistry is of crucial importance. A better understanding of TTM pathogenesis is essential to improve diagnosis and treatment.

## Figures and Tables

**Figure 1 diagnostics-14-00553-f001:**
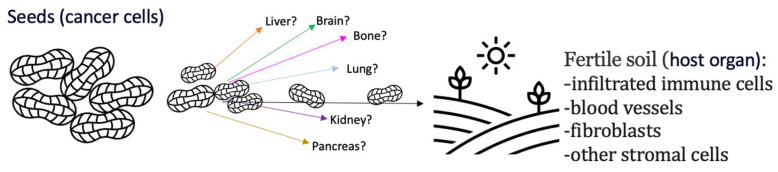
The seed and soil hypothesis: The ability of cancer to metastasize depends on the microenvironment of the target tissue.

**Figure 2 diagnostics-14-00553-f002:**
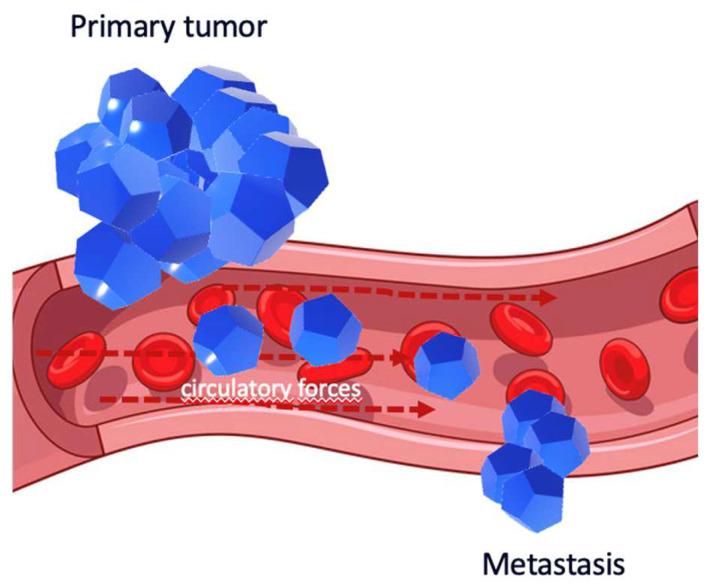
The mechanical theory: it is probable that metastasis will happen in areas that correspond to the pattern of blood flow.

**Figure 3 diagnostics-14-00553-f003:**
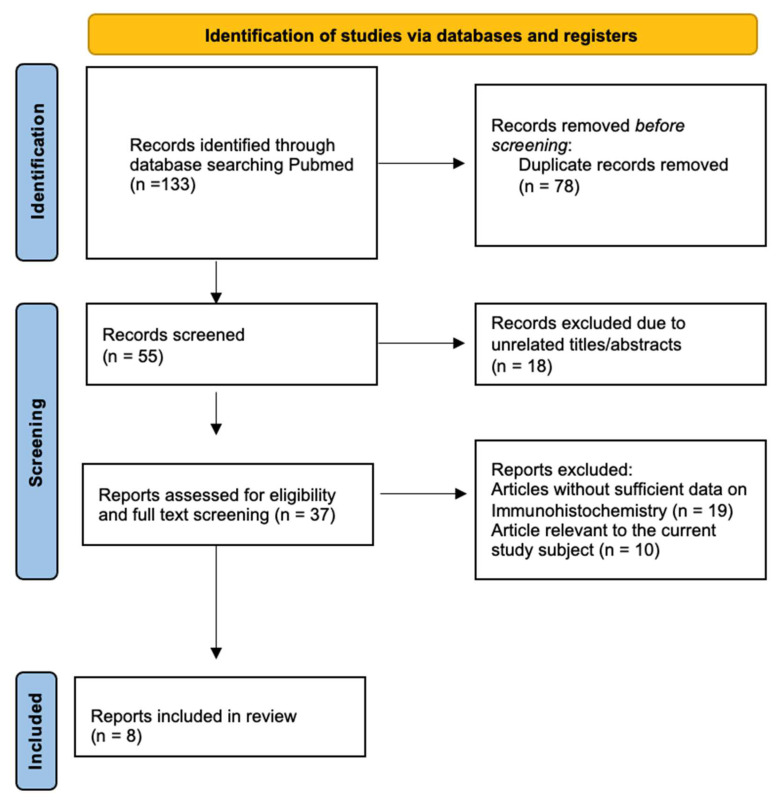
PRISMA flowchart of the search process.

**Figure 4 diagnostics-14-00553-f004:**
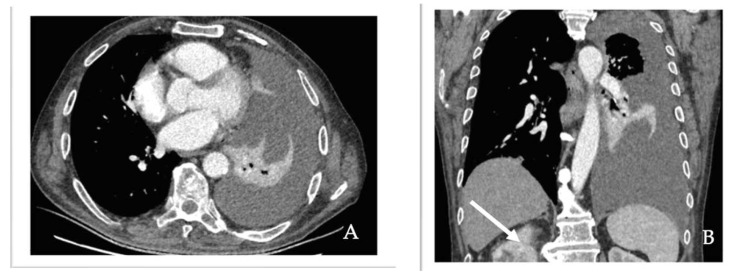
(**A**) Chest CT scan showing in axial section left-sided massive pleural effusion. (**B**) Incidentally discovered right kidney lesion (arrow) on coronal section of the chest CT scan.

**Figure 5 diagnostics-14-00553-f005:**
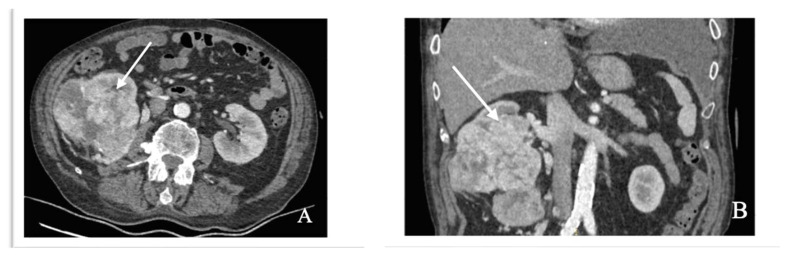
Contrast-enhanced abdominal CT images demonstrates an enhancing mass (arrow) in the right kidney with areas of hemorrhage. (**A**) Axial section. (**B**) Coronal section. We obtained ethical approval for the CT scan images used in the study.

**Figure 6 diagnostics-14-00553-f006:**
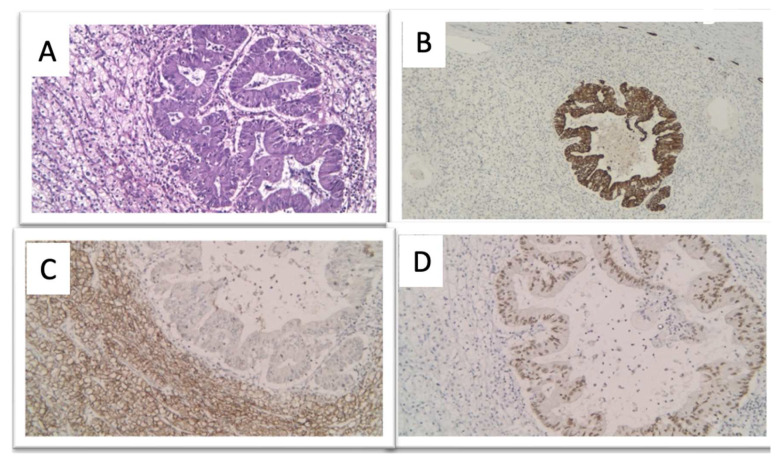
(**A**) The RCC intermingled with an adenocarcinoma morphologically similar to the lung adenocarcinoma (H&E, 20×). (**B**) CK7 immunoreactivity in the lung ADK. (**C**) TTF1 positivity in lung ADK. (**D**) CAIX positivity (immunoperoxidate 20×). We obtained ethical approval for the IHC staining used in the study.

**Table 1 diagnostics-14-00553-t001:** Criteria to diagnose TTM.

Campbell’s proposed criteria	The presence of more than two primary cancers The host or the recipient tumor must be a true benign or malignant neoplasm, with the exception of lymphomas or leukemia The metastatic or donor tumor must develop in juxtaposition to the recipient tumor with areas of intermingling
Han’s proposed criteria	The metastatic lesion must be partially surrounded by the host tumor tissue The metastasis must be proven to have originated from another primary cancer
Richter’s proposed criteria	The presence of tumor foci with distinct histology in the host tumor The presence of a metastasized primary cancer

**Table 2 diagnostics-14-00553-t002:** Summary of 12 reported cases of tumor-to-tumor metastasis from lung cancer to kidney cancer in the English literature (M = male, F = female, HP = histopathology, IHC = immunohistochemistry, SCLC = small-cell lung carcinoma, ADK = adenocarcinoma, M1HEP = hepatic metastases, M1LYM = lymphatic metastasis, M1OSS = bone metastasis, M1PUL = lung metastasis, CAIX = carbonic anhydrase IX, Vim = vimentin, TTF-1 = thyroid transcription factor-1, CK7 = cytokeratin 7).

Study	Year	Demographic Information		Histology Diagnosis						Other Visceral Metastasis
		Gender	Age	Donor Tumor			Receptor Tumor			
				HP	IHC	Size (cm)	HP	IHC	Size (cm)	
Matsukuma S. et al. [91]	2013	M	88	Lung ADK	Napsin A+ CK7+ Vim+	8.2	Clear-cell RCC	Vim+ CK7− PAX8	1.3	M1PUL
		M	69	Lung ADK	Napsin A+ CK7+	2.5	Clear-cell RCC	CA IX+ CK7−	3	-
		M	72	Lung ADK	TTF-1+ CK7+	6.5	Papillary RCC	CK7+, AMACR+ CAIX−	8	M1HEP
		M	48	Lung ADK	Napsin A+ CK7+	2.6	Clear-cell RCC	CA IX+ CK7−	1.2	M1OSS
		M	82	SCLC	Vim+ TTF-1+ Synatopsin+ CD56+	6	Clear-cell RCC	CA IX+ CK7−	2	-
Sawada T et al. [92]	2009	F	97	Lung ADK	TTF-1+ CK7+	4	Clear-cell RCC	CA IX+ CK7−	4.4	-
Rais G, et al [93]	2022	M	52	Squamous cell lung cancer	P40+	7.2	Clear-cell RCC	Vim+, CD10+	3	-
Wu PS, et al. [94]	2015	F	72	Lung ADK	TTF-1+ CK7+	5	Renal angiomyo-lipoma	smooth muscle actin and HMB- 45	10	-
Altinok G, et al. [95]	1999	M	64	Squamous cell lung cancer	CK5/6 +	4.5	Renal oncocitoma	CD117+ E-cadherin+	6.5	-
Duprez R, et al. [96]	2009	M	60	SCLC	Vim− CD10−, CK 7+ TTF1+, Chromogranin+ CD56+	2.5	Clear-cell RCC	Vim+ CD10+ CK 7−	2.5	M1LYM, M1HEP
Sella A, et al. [97]	1987	F	56	Lung ADK	TTF-1+ CK7+	5.5	Clear-cell RCC	CA IX+ CK7−	7	M1LYM M1OSS M1HEP
Granville LA, et al. [98]	2005	F	65	Lung ADK	TTF-1+ CK7+	1	Clear-cell RCC	Vim+ CK 7−	4	M1OSS

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
