# Peer review of "Tumor-to-Tumor Metastasis of Lung Cancer to Kidney Cancer: A Review of the Literature and Our Experience"

_diagnostics, 2024, doi:10.3390/diagnostics14050553_

Round 1
Reviewer 1 Report
Comments and Suggestions for Authors
The paper, titled "Tumour-to-Tumour Metastasis of Lung Cancer to Kidney Cancer: A Review of the Literature and Our Experience," discusses a rare phenomenon known as tumour-to-tumour metastasis (TTM), where metastasis occurs between two distinct primary tumors. In this review, the authors focus on cases where lung cancer metastasizes to renal cell carcinoma (RCC). They outline the proposed models and mechanisms involved in TTM, summarize, and analyze previous case reports, and present their own experience with a case of lung cancer metastasizing into RCC. The paper emphasizes the importance of considering TTM in differential diagnosis and suggests that a better understanding of its pathogenesis could improve diagnosis and treatment strategies. The topic is interesting, and this manuscript is well-written. However, there are still several concerns and comments need to be addressed before considering publication.
1.The abstract requires reorganization as it currently includes detailed analysis processes, such as those described in lines 18-20. These details should be included in the main text rather than in the abstract.
2.While the paper's literature review on tumor-to-tumor metastasis is comprehensive, limitations such as database and keyword constraints, potential language biases, date range limitations, selection criteria, publication bias, and the inherent challenges of manual search methods may have restricted the inclusion of all relevant studies.
3.The figure legends currently lack detail and would benefit greatly from further enrichment and elaboration by the author.
4.Could the authors specify whether the CT scan images and IHC staining of the cancer patients are sourced from published databases or literature? It would be beneficial if this is clearly stated in the main text. Additionally, if these images were produced by the authors, it is essential to mention the ethical approval obtained for their use.
5.In Figures 4, 5, and 6, could the authors please include scale bars to provide a reference for the size of the depicted elements?
Comments on the Quality of English LanguageMinor editing of English language required
Author Response
Authors’ response to reviewer’s 1 comments
We thank the reviewer again for the close reading of our article and all the constructive and detailed comments.
Comment 1: The abstract requires reorganization as it currently includes detailed analysis processes, such as those described in lines 18-20. These details should be included in the main text rather than in the abstract.
Answer: We greatly appreciate the reviewer for bringing this to our attention. We have removed the mentioned details from the abstract.
Comment 2: While the paper's literature review on tumor-to-tumor metastasis is comprehensive, limitations such as database and keyword constraints, potential language biases, date range limitations, selection criteria, publication bias, and the inherent challenges of manual search methods may have restricted the inclusion of all relevant studies.
Answer: We acknowledge the significant limitations mentioned by the reviewer and have included them in our manuscript.
Comment 3: The figure legends currently lack detail and would benefit greatly from further enrichment and elaboration by the author.
Answer: We thank the reviewer very much for the reminder. We have made revisions accordingly.
Comment 4: Could the authors specify whether the CT scan images and IHC staining of the cancer patients are sourced from published databases or literature? It would be beneficial if this is clearly stated in the main text. Additionally, if these images were produced by the authors, it is essential to mention the ethical approval obtained for their use.
Answer: We thank the reviewer for raising this important issue.The CT scan images and IHC staining used in the study were not obtained from published databases or literature, but were produced by the authors themselves. In response to the reviewer's suggestion, we have made it clear in the main text that we obtained ethical approval for the use of these materials.
Comment 5: In Figures 4, 5, and 6, could the authors please include scale bars to provide a reference for the size of the depicted elements?
Answer: We are very grateful to the reviewer for the reminder provided. We have made revisions accordingly.
We sincerely thank the reviewer again for providing constructive feedback to improve our manuscript!

Reviewer 2 Report
Comments and Suggestions for Authors
1. lns 29 to 39 are too generic and long winded; so please rewrite to up to 3 sentence-single paragraph of a pithy and informative “introduction”
2. Acronyms/Abbreviations/Initialisms should be defined the first time they appear in each of three sections: the abstract; the main text; the first figure or table (CAIX and RCC from figure 6).
3. Figure 3 should be PRISMA diagram, following the PRISMA statement since only 12 case reports are finally selected, authors could easy stipulate a #ridk of bias# table.
4. lns 91 to 93 . a sentence cannot be a separate paragraph!
5. conclusions are equally long winded as the “introduction”; limit to 4-4 sentences, without repeating the discussion.
6. wort on the plagiarism!
Author Response
Authors’ response to reviewer’s 1 comments
We thank the reviewer again for the close reading of our article and all the constructive and detailed comments.
Comment 1: lns 29 to 39 are too generic and long winded; so please rewrite to up to 3 sentence-single paragraph of a pithy and informative “introduction”
Answer: We greatly appreciate the reviewer for bringing this to our attention. We have made revisions accordingly.
Comment 2: Acronyms/Abbreviations/Initialisms should be defined the first time they appear in each of three sections: the abstract; the main text; the first figure or table (CAIX and RCC from figure 6).
Answer: We thank the reviewer very much for the reminder. We have made revisions accordingly.
Comment 3: Figure 3 should be PRISMA diagram, following the PRISMA statement since only 12 case reports are finally selected, authors could easy stipulate a #ridk of bias# table.
Answer: We thank the reviewer very much for the reminder. We have made revisions accordingly.
Comment 4: lns 91 to 93 . a sentence cannot be a separate paragraph!
Answer: We thank the reviewer for raising this important issue. We have made the necessary revisions accordingly.
Comment 5: Conclusions are equally long winded as the “introduction”; limit to 4-4 sentences, without repeating the discussion.
Answer: We appreciate the reviewer's reminder and have made revisions accordingly.
Comment 5: wort on the plagiarism!
Answer: We thank the reviewer for raising this important issue. We ckecked the plagiarism
We sincerely thank the reviewer again for providing constructive feedback to improve our manuscript!

Round 2
Reviewer 2 Report
Comments and Suggestions for Authors
Thank you for appraising almost all of my comments. Whatsoever, in order to bw regarded highly, this article still misses
a) explicit yet concise methodology with respect to tull-text obtaining (automated software? online repositories?)
b) assessed studies should be listed in the PICO-based table
c) All the studies included in that table should be evaluated for different types of biases. See the examples in https://methods.cochrane.org/bias/resources/rob-2-revised-cochrane-risk-bias-tool-randomized-trials – there is no need to make a table of biases – this is just an illustrative example.
Author Response
Authors’ response to reviewer’s 2 comments
We appreciate your feedback and suggestions on our manuscript. Based on your comments, we have made appropriate revisions to the paper. Thank you for your time and input.
Comment 1: explicit yet concise methodology with respect to tull-text obtaining (automated software? online repositories?)
Answer: We thank the reviewer for the constructive remarks. We have revised it accordingly.
In order to identify studies that report lung cancer metastasis to RCC, we conducted an extensive literature search using the best available resources. We scoured the online repositories of PubMed-MEDLINE from their inception to December 2023. Our search query line was carefully crafted to ensure we found every relevant piece of literature. The line was as follows: (("tumour-to-tumour metastasis" [All Fields]) OR ("kidney cancer") OR ("lung cancer" [All Fields])).
Comment 2: assessed studies should be listed in the PICO-based table
Answer: We appreciate the reviewer's reminder. We attempted to improve the table, but it did not fully comply with PICO criteria as the data was mainly obtained from autopsy cases and focused on immunohistochemical findings. The research question is how to properly diagnose TTM. The intervention involved using diagnostic tests such as IHC stains, with the goal of achieving a correct histological diagnosis.
Comment 3: All the studies included in that table should be evaluated for different types of biases. See the examples in https://methods.cochrane.org/bias/resources/rob-2-revised-cochrane-risk-bias-tool-randomized-trials – there is no need to make a table of biases – this is just an illustrative example.
Answer: We fully acknowledge that case reports and case series carry a higher risk of bias. It's important to note that medical records may not always contain all relevant data, which can inadvertently impact the quality and interpretation of our observations due to our own subjectivity.
Thank you again for your thorough and insightful review. We hope that the revised manuscript meets your satisfaction and look forward to hearing back from you!

Round 3
Reviewer 2 Report
Comments and Suggestions for Authors
No further comments